# Episodic Memory Representation for Long-form Video Understanding

## Abstract

Video Large Language Models (Video-LLMs) excel at general video understanding but struggle with long-form videos due to context-window limits. Consequently, recent approaches focus on keyframe retrieval, condensing lengthy videos into a small set of informative frames. Despite their practicality, these methods simplify the problem to static text-image matching, overlooking spatio-temporal relationships crucial for capturing scene transitions and contextual continuity, and may yield redundant keyframes with limited information, diluting salient cues essential for accurate video question answering. To address these limitations, we introduce Video-EM, a training-free framework inspired by the principles of human episodic memory, designed to facilitate robust and contextually grounded reasoning. Rather than treating keyframes as isolated visual entities, Video-EM represents them as temporally ordered episodic events, capturing both spatial relationships and temporal dynamics essential for reconstructing the underlying narrative. Furthermore, it integrates Chain-of-Thought (CoT) reasoning to iteratively identify a minimal yet highly informative subset of episodic memories, enabling efficient and accurate question answering by Video-LLMs. Extensive evaluations on multiple mainstream long-video benchmarks demonstrate the superiority of Video-EM, which achieves highly competitive results while using fewer frames.

## 1 Introduction

The rapid advancement of Video Large Language Models (Video-LLMs) has achieved remarkable progress in video understanding (Xu et al., 2021; Zhang et al., 2025b), particularly in video question answering Zhang et al. (2023b), demonstrating strong potential for modeling real-world scenarios Song et al. (2025); Zhang et al. (2025a). However, as video content moves from minutes to hours-long sequences, the limited context window of Video-LLMs poses a critical bottleneck for long-form video understanding.

To bridge this gap, recent research on long-form video understanding has emphasized training-free frame sampling strategies Liu et al. (2025); Yu et al. (2024); Tang et al. (2025); Wang et al. (2025). While effective, these methods typically simplify long-video understanding as static text–image matching by selecting query-relevant keyframes, resulting in two primary limitations. First, methods like Wang et al. (2025) process keyframes to per-frame captions in Video-LLMs, overlooking temporal dependencies essential for explicitly capturing scene transitions and contextual continuity in complex narratives. Second, query-driven sampling tends to yield visually redundant frames in long videos. Such redundancy may dilute the salient cues essential for accurate question answering, and increase computational costs while underutilizing the limited context window in Video-LLMs Sun et al. (2025), as illustrated in Figure 1.

To mitigate these limitations, we introduce Video-EM, a training-free framework inspired by human cognitive mechanisms. Humans excel at lifelong reasoning by encoding salient experiences as hierarchical episodic memories [1], each anchored in specific spatio-temporal contexts Rothfuss et al. (2018); Hampton & Schwartz (2004). Drawing on this insight, Video-EM reformulates long-form video question answering by treating isolated keyframes as temporally ordered key events. These

---

[1] In humans, episodic memory refers to the capacity to form, store, and consciously retrieve specific past events within their unique spatio-temporal contexts, effectively enabling "mentally time travel" to reexperience those moments.

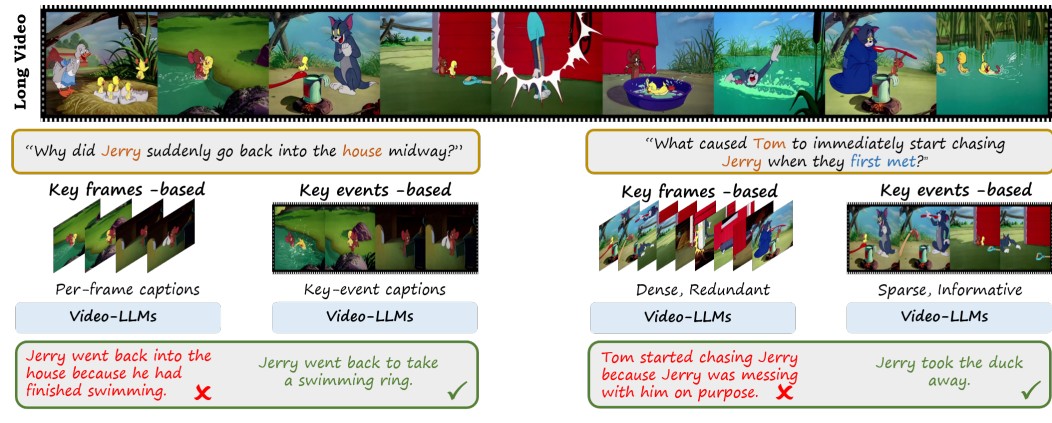

Figure 1: Illustration of the limitation of existing training-free sampling methods. (a) Treating frames as isolated instances results in temporal discontinuities that disrupt the semantic narrative of events. (b) Densely redundant frames with limited information can occupy the model's limited context window, diluting key cues and compromising overall performance.

events are transformed into episodic memory representations that capture both intricate spatial scene relationships and temporal dynamics, enabling robust and contextually grounded reasoning. Next, the framework leverages chain-of-thought (CoT) thinking with Large Language Models (LLMs) to evaluate whether the current episodic memory provides sufficient information for reliable video comprehension. Through this iterative reasoning process, the framework retrieves the minimal yet highly informative subset of episodic memories for subsequent answering by Video-LLMs, ensuring both efficiency and accuracy.

Concretely, given a video-based query, Video-EM performs multi-grained semantic retrieval by matching CLIP embeddings of decomposed query tokens against frame embeddings. Rather than treating retrieved keyframes as isolated inputs, Video-EM orders them temporally as key events and applies adaptive event expansion to enhance semantic coherence, mitigating the risk of missing diverse contextual information essential for answering complex queries. It then encodes these expanded events into structured episodic memories by employing object detection and video captioning modules, extracting both scene relationships and dynamic event narratives. This allows the existing Video-LLMs to "replay" each moment with explicit grounding of *when* it happened, *where* it took place, *what* occurred, and *which* objects were involved. Finally, Video-EM employs a CoT strategy to iteratively reason over episodic memories and select the most relevant subset. For finer analysis, it recursively decomposes events into shorter temporal segments. This process yields a minimal yet highly informative memory set, enabling the downstream Video-LLM to answer the query accurately.

In summary, our contributions are threefold: (1) We propose a novel paradigm for long-form video understanding as episodic memory retrieval and reasoning, moving beyond frame-centric methods to a human-inspired memory mechanism that captures spatio-temporal context for robust and contextually grounded reasoning. (2) We propose Video-EM, a training-free framework that integrates an episodic memory module with a chain-of-thought (CoT) thinking strategy. It efficiently retrieves a minimal yet informative subset of memories for Video-LLMs, enhancing both accuracy and efficiency. (3) Experiments on long video understanding benchmarks demonstrate that Video-EM outperforms prior methods and is broadly compatible with mainstream Video-LLMs, achieving consistent improvements.

## 2 RELATED WORK

### 2.1 LONG-FORM VIDEO UNDERSTANDING

Long-video understanding is inherently more challenging than those faced in short-video or image-based tasks due to its complex temporal dynamics and significant visual redundancy Qian et al.

(2024); Zeng et al. (2024); Faure et al. (2024). While several methods, such as extended context windows Chen et al. (2024); Zhang et al. (2024a); Team et al. (2023) and token compression Li et al. (2024); Weng et al. (2024); Song et al. (2023), have attempted to address these issues, they often struggle to balance computational efficiency with performance, particularly when handling hours-long content. To mitigate these limitations, recent advances in query-aware keyframe sampling Tang et al. (2025); Yu et al. (2024); Liu et al. (2025) have been developed to aim to alleviate redundancy by selecting frames relevant to queries. Parallel efforts have extended Chain-of-Thought (CoT) thinking to Video-LLMs Fei et al. (2024); Zhang et al. (2023a); Wang et al. (2024c), leveraging LLMs as agents to interpret visual content through enhanced reasoning capabilities. However, these approaches tend to process keyframes independently, lacking mechanisms to explicitly model temporal dependencies or evolving narratives. A representative work, VideoTree Wang et al. (2025) employs a CoT-based strategy to retrieve keyframes and generate frame-wise captions in isolation, but this approach overlooks the causal and temporal relationships necessary for coherent video understanding. To advance long-video understanding, we argue that it is essential to move beyond isolated frame processing toward temporally grounded reasoning. This requires capturing both fine-grained visual semantics and high-level narrative progression. By structuring keyframes into coherent spatio-temporal representations, we empower Video-LLMs to perform more effective spatial and temporal reasoning over long-form videos.

## 2.2 Episodic Memory for Vision-Language Models

Humans excel at lifelong reasoning by leveraging episodic memory, which hierarchically encodes salient personal experiences within specific spatio-temporal contexts Lee & Dey (2007); Moscovitch et al. (2016); Fountas et al. (2025). This cognitive mechanism has been introduced to robotics, where episodic memory can be conceptualized as hierarchical text descriptions generated by vision-language models (VLMs). This enables embodied agents to encode experiences over time for effective decision-making and action planning Rothfuss et al. (2018); Bärmann et al. (2021). Among recent advancements, the work most related to ours is LifelongMemory Wang et al. (2024d). It introduces a text-based episodic memory framework that leverages video narrations for downstream video understanding tasks. While this approach excels at identifying activities (the 'what'), it largely overlooks the complex spatio-temporal context (the 'when', 'where', and 'which objects'), which are critical for effective spatio-temporal reasoning. This limitation is evident in the V-STaR benchmark Cheng et al. (2025), which demonstrates that many models excel at identifying actions ('what') but struggle to accurately ground them in time and space. In contrast, Video-EM directly models these spatio-temporal relationships by meticulously capturing both fine-grained visual scene relationships and the dynamics of evolving events. These enriched representations form a compact yet informative episodic memory, thereby facilitating robust comprehension for long-form videos.

## 3 Methodology

We introduce Video-EM, a training-free pipeline designed for seamless integration with any Video-LLMs. As depicted in Figure 2, Video-EM comprises three core components: Key Event Selection, Episodic Memory Representation, and CoT Video Reasoning.

To address the inherent redundancy in long videos, the Key Event Selection module first isolates query-relevant content by extracting temporally contiguous key events. To capture temporal dynamics and contextual relationships beyond what similarity-based retrieval can expose, the Episodic Memory Representation module then adaptively expands these segments to recover complete events. It applies object detection and captioning to derive dynamic scene relationships (e.g., who interacts with what, and where) and dynamic scene narratives, forming a comprehensive episodic memory that records the *when*, *where*, *what*, and *which objects* of each event—effectively 'replaying' the video's critical moments. Finally, Video-EM employs a chain-of-thought (CoT) strategy to iteratively select a minimal yet highly informative subset of episodic memories, refining them into finer-grained representations when needed to produce precise, contextually grounded answers.

### 3.1 Key Event Selection

Naïve query-relevant keyframe selection often suffers from coarse-grained semantic matching, leading to the loss of critical contextual information due to inadequate semantic coverage, particularly in

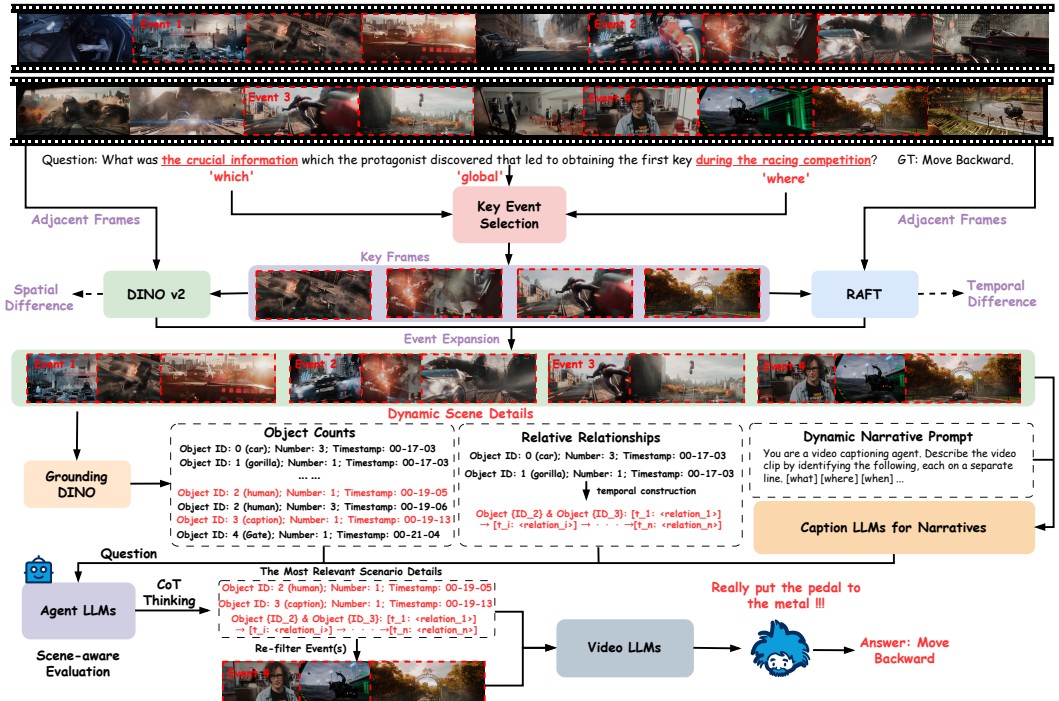

Figure 2: The pipeline of the Video-EM framework consists of three steps: Key Event Selection, Episodic Memory (EM) Representation, and CoT-based video reasoning.

long videos Ma et al. (2022). To mitigate this, our Key Event Selection module first conducts multi-grained semantic retrieval, followed by key event segmentation, which groups temporally adjacent and semantically related frames into coherent events, thereby enabling more robust modeling of dynamic temporal relationships.

**Multi-grained Semantic Retrieval.** Given a video $V = \{f_i\}_{i=1}^N$ consisting of $N$ frames, where $f_i$ is the $i$-th frame, our objective is to extract a set of representative keyframes that are most relevant to a given natural language query $q$. To achieve this, we first decompose $q$ into a multi-granular query set $Q = \{q, q_o, q_s\}$, where $q$ is the original query, $q_o$ captures object-level semantics (e.g., "apple"), and $q_s$ captures scene-level context (e.g., "kitchen"), enabling robust and semantically aligned retrieval. For each candidate frame $f_i$, we compute its similarity score as follows:

$$Sim(f_i) = \sum_{q_i \in Q} \omega_{q_i} \phi_I(f_i) \cdot \phi_T(q_i), \tag{1}$$

where $\phi_I$ and $\phi_T$ denote the CLIP image and text encoders, with weights $\omega_{q_1}$ for the original query and $\omega_{q_2} = \omega_{q_3}$ for the query components. Finally, we select the top $K$ frames with the highest similarity scores and arrange them in temporal order to form the keyframe set $V^* = \{f_i\}_{i=1}^K$.

**Temporal Event Segmentation.** To ensure temporal coherence in event grouping, we partition the set of keyframes $V^* = \{f_i\}_{i=1}^K$ into distinct events according to their timestamps. Specifically, we define event boundaries by enforcing a minimum temporal gap of $\Delta t$ between consecutive events. Keyframes whose timestamps differ by less than $\Delta t$ from their neighbors are grouped into the same event $E_i$. Formally, each event $E_i$ consists of a sequence of frames $\left\{ f_j^{E_i} \right\}_{j=1}^{n_i}$, where $n_i \leq N$ is the number of frames in $E_i$. The temporal distance between the last frame of $E_i$ and the first frame of $E_{i+1}$ must satisfy $|t_1^{(E_{i+1})} - t_{n_i}^{(E_i)}| > \Delta t$. In our implementation, we empirically set $\Delta t$ to 3. By iteratively applying this procedure to all keyframes, we obtain a set of temporally coherent events $\mathbf{E} = \{E_1, E_2, \ldots, E_M\}$, where $M \leq K$.

## 3.2 EPISODIC MEMORY REPRESENTATION

After obtaining the key event set $E$, we adaptively expand each clip to recover query-relevant context that similarity-based filtering may miss, thereby mitigating the limitations of purely semantic retrieval and preserving subtle transitions and causal cues. The expanded events are then encoded as episodic memory representations—drawing on insights from cognitive science Holland & Smulders (2011); Hampton & Schwartz (2004)—that explicitly capture the *when*, *where*, *what*, and *which objects* involved in each event, enabling rich modeling of dynamic spatio-temporal context.

**Adaptive Event Expansion.** To enhance the semantic coherence of the key event set $E$, we introduce an adaptive event expansion mechanism driven by a spatio-temporal difference metric. This mechanism compensates for potentially omitted yet contextually relevant frames during keyframe selection by explicitly analyzing frames near event boundaries. For the key event $\{f_j^{E_i}\}_{j=1}^{n_i}$, we compute the spatio-temporal difference between its boundary frames and neighboring frames outside the event. The spatial difference captures appearance variation via L1 distance between DINOv2 Oquab et al. (2023) visual embeddings: $D_{\text{spatial}} = |\Phi(f_i) - \Phi(f_{i-1})|_1$, where $\Phi$ denotes the frame-level embedding. and $|\cdot|_1$ represents L1-normalization. The temporal difference is estimated from optical flow displacement, with the average motion magnitude computed as $D_{\text{flow}}(f_i) = \frac{1}{HW} \sum_{x,y} \sqrt{u_i(x,y)^2 + v_i(x,y)^2}$, where $F_i = (u_i, v_i)$ is the flow between $f_i$ and $f_{i-1}$ calculated by RAFT Teed & Deng (2020). The spatio-temporal difference score $S_i$ is obtained by combining these components:

$$S_i = \alpha \cdot D_{\text{spatial}}(f_i) + (1 - \alpha) \cdot D_{\text{flow}}(f_i), \tag{2}$$

where $\alpha \in [0, 1]$ is a tunable hyperparameter balancing spatial and temporal contributions. Starting from event boundaries, adjacent frames are iteratively added to the event, until $S_i > \tau_{st}$ or a predefined limit $L_{\max}$ is reached. In our implementation, we empirically set $\alpha = 2$, $\tau_{st} = 2$, and $L_{\max} = 3$, ensuring bounded yet semantically enriched expansion.

**Dynamic Scene Narratives.** To comprehensively encode the *when*, *where*, and *what* of each event $E_i$, we employ a state-of-the-art video captioning model Yuan et al. (2025a), enhanced with a custom-designed thinking strategy (see Appendix). This model generates hierarchical and semantically rich summaries that are tightly anchored to temporal structure. In contrast to prior per-frame captioning methods—often producing redundant or fragmented descriptions Wang et al. (2025), our approach delivers coherent, clip-level narratives that effectively capture temporal evolution. Each summary $\mathbf{N}^{scene}$ explicitly encodes the temporal position of the event (*when*), its spatial context (*where*), and the core actions and entities involved (*what*). Beyond these scene-level summaries, complex video understanding further demands modeling the dynamic interactions and evolving relationships among objects, which are critical for achieving deeper situational comprehension and causal reasoning.

**Dynamic Scene Relationships.** To address this, we construct event-centric scene representations that go beyond isolated object descriptions. Specifically, to explicitly capture the dynamic relationships of "*which objects*" evolve and interact throughout each event, we encode fine-grained spatial and relational structures, denoted as $\mathbf{G}^{scene}$. Leveraging a robust object detection framework Liu et al. (2024), we identify salient objects to model their dynamic spatial properties through two core components: *(i) Evolution of Object Counts* $\mathbf{A}_{cnt}$: This component characterizes the temporal evolution of object quantities by detecting their appearance and disappearance over time. It represents these dynamics in the following format: "The count of Object $\{ID_1\}$ evolves as follows: $\{num_1\}$ at $t_1$, $\{num_2\}$ at $t_2$, ..., $\{num_n\}$ at $t_n$." *(ii) Evolution of Location Relationships* $\mathbf{A}_{loc}$: Captures temporal changes in pairwise spatial locations between objects, formatted as: " Location evolution for Object $\{ID_1\}$ & Object $\{ID_2\}$: $[t_1$: <relation$_1$>] $\rightarrow$ $[t_i$: <relation$_i$>] $\rightarrow \cdots \rightarrow [t_n$: <relation$_n$>]." These components combine into a structured scene relatiqonship representation: $\mathbf{G}^{scene} = \{\mathbf{A}_{cnt}, \mathbf{A}_{loc}\}$, which provides an interpretable and structured representation of evolving scenes for video reasoning. In this way, the fine-grained spatio-temporal relationships captured by $\mathbf{G}^{scene}$ and the high-level narratives in $\mathbf{N}^{scene}$ complement each other, together forming a rich spatio-temporal episodic memory that supports effective, context-aware video reasoning.

## 3.3 CHAIN-OF-THOUGHT VIDEO REASONING

Rather than providing the entire episodic memory to the subsequent Video-LLMs, which risks diluting relevant information with excessive context Liu et al. (2025), we employ a CoT thinking strategy with an LLM as an agent to iteratively retrieve a minimal yet highly relevant subset of episodic memories.

Table 1: Quantitative comparison on four popular long-form video benchmarks. We use Qwen2-VL and Qwen2.5-VL as baselines to showcase our video-EM. For baseline methods, we reproduce results on benchmarks using official code and mark them with ‡.

**Video-MME**

| Model | Venue | Size | Frames | Overall 17 min | Short 1.3 min | Medium 9 min | Long 41 min |
|---|---|---|---|---|---|---|---|
| *Proprietary Models* | | | | | | | |
| GPT-4V | - | - | 256 | 59.9 | - | - | - |
| GPT-4o | - | - | 32 | 62.5 | 71.4 | 61.0 | 55.2 |
| *Open-source Models* | | | | | | | |
| Video-LLaVA | ArXiv'23 | 7B | 8 | 41.6 | 46.1 | 40.7 | 38.1 |
| VideoLLaMA2 | ArXiv'24 | 7B | 8 | 47.9 | 56.0 | 45.4 | 42.1 |
| LongVA | ArXiv'24 | 7B | 128 | 52.6 | 61.1 | 50.4 | 46.2 |
| ShareGPT4Video | NIPS'24 | 8B | 16 | 43.6 | 53.6 | 39.3 | 37.9 |
| LongVU | ICML'25 | 7B | 1fps | 60.6 | 64.7 | 58.2 | **59.5** |
| Video-XL | CVPR'25 | 7B | 128 | 55.5 | 64.0 | 53.2 | 49.2 |
| VideoChat2 | CVPR'24 | 7B | 16 | 43.8 | 52.8 | 39.4 | 39.2 |
| LLaVA-OneVision | TMLR'24 | 7B | 32 | 58.2 | 69.1 | 53.3 | 46.7 |
| Frame-Voyager | ICLR'25 | 8B | - | 57.5 | 67.3 | 56.3 | 48.9 |
| *Training-free Models* | | | | | | | |
| VideoTree | CVPR'25 | - | 128 | 54.2 | - | - | - |
| ∞-VideoChat2 | ICML'25 | 7B | 128 | 42.4 | 48.1 | 40.2 | 38.9 |
| Qwen2-VL w/ AKS | CVPR'25 | 7B | 32 | 59.9 | - | - | - |
| Qwen2-VL w/ Q-Frame | ICCV'25 | 7B | 44 | 58.3 | 69.4 | 57.1 | 48.3 |
| Qwen2-VL‡ | ArXiv'24 | 7B | 32 | 56.9 | 68.7 | 53.3 | 48.8 |
| **Qwen2-VL w/ Ours** | - | 7B | Avg 28 | 60.6 | 70.7 | 58.8 | 52.3 |
| Qwen2.5-VL‡ | ArXiv'25 | 7B | 32 | 59.2 | 70.6 | 56.7 | 50.6 |
| **Qwen2.5-VL w/ Ours** | - | 7B | Avg 28 | **62.0** | **72.4** | **60.3** | 53.4 |

**LVBench**

| Model | Venue | Size | Frames | Overall 68 min |
|---|---|---|---|---|
| GPT-4o | - | - | - | 34.7 |
| Gemini 1.5 pro | - | - | - | 33.1 |
| LLaVA-Video | ArXiv'24 | 7B | 180 | 41.5 |
| VideoLLaMA2 | ArXiv'24 | 7B | 180 | 36.2 |
| LongVU | ICML'25 | 7B | 1fps | 37.8 |
| Video-XL | CVPR'25 | 7B | - | 36.8 |
| LongLLaVA | ArXiv'24 | 9B | - | 31.2 |
| VAMBA | CVPR'25 | 10B | 1024 | 42.1 |
| Qwen2-VL‡ | ArXiv'24 | 7B | 32 | 38.4 |
| **Qwen2-VL w/ Ours** | - | 7B | Avg 27 | 45.2 |
| Qwen2.5-VL‡ | ArXiv'25 | 7B | 32 | 36.6 |
| **Qwen2.5-VL w/ Ours** | - | 7B | Avg 27 | **45.7** |

**HourVideo**

| Model | Venue | Size | Frames | Overall 47 min |
|---|---|---|---|---|
| GPT-4 | - | - | 0.5fps | 25.7 |
| Gemini 1.5 Pro | - | - | 0.5fps | 37.3 |
| LongVU | ICML'25 | 7B | 1fps | 30.8 |
| VideoLLaMA3 | ArXiv'25 | 7B | 32 | 31.0 |
| VAMBA | CVPR'25 | 10B | 128 | 33.6 |
| Qwen2-VL‡ | ArXiv'24 | 7B | 32 | 30.7 |
| **Qwen2-VL w/ Ours** | - | 7B | Avg 30 | 34.8 |
| Qwen2.5-VL‡ | ArXiv'25 | 7B | 32 | 31.1 |
| **Qwen2.5-VL w/ Ours** | - | 7B | Avg 30 | **35.1** |

When finer-grained analysis is required, the CoT framework adaptively decomposes the selected event into temporally refined episodic memory representations through visual clustering, allowing the model to capture subtle transitions and nuanced contextual cues that might otherwise be overlooked. After identifying these critical episodic memories, they are integrated with both the input query and the corresponding event to construct a composite input, which is subsequently fed into the Video-LLM for final inference. This design not only enhances the model's ability to reason over complex video narratives but also ensures that the inference process remains efficient and precise. A detailed description of the complete CoT prompting strategy can be found in the Appendix.

Table 2: Comparison with SoTA methods on EgoSchema.

| Model | Venue | Size | Frames | Overall 3 min |
|---|---|---|---|---|
| PLLaVA | ArXiv'24 | 7B | 16 | 45.6 |
| VideoLLaMA2 | ArXiv'24 | 7B | 8 | 53.1 |
| VideoChat2 | CVPR'24 | 7B | 16 | 54.4 |
| LLaVA-OneVision | TMLR'24 | 7B | 32 | 60.1 |
| VideoLLaVA | EMNLP'24 | 7B | 8 | 40.2 |
| LLoVi | EMNLP'24 | GPT-3.5 | 90 | 57.6 |
| VideoAgent | ECCV'24 | GPT-4 | 8.4 | 60.2 |
| VideoLLaMa | ICCV'25 | 7B | 8 | 53.8 |
| VideoTree | CVPR'25 | GPT-4 | 63.2 | **66.2** |
| BOLT | CVPR'25 | 7B | 16 | 61.8 |
| MVU | ICLR'25 | 7B | 16 | 60.2 |
| Qwen2-VL‡ | ArXiv'24 | 7B | 8 | 59.6 |
| Qwen2-VL‡ | ArXiv'24 | 7B | 16 | 61.2 |
| **Qwen2-VL w/ Ours** | - | 7B | Avg 9 | 65.6 |
| Qwen2.5-VL‡ | ArXiv'25 | 7B | 8 | 56.4 |
| Qwen2.5-VL‡ | ArXiv'25 | 7B | 16 | 60.2 |
| **Qwen2.5-VL w/ Ours** | - | 7B | Avg 9 | 64.4 |

## 4 EXPERIMENTS

### 4.1 EXPERIMENTAL SETUP

**Benchmarks.** We evaluate the performance of Video-EM in four popular benchmarks: 1) **Video-MME** Fu et al. (2025), comprising 2700 question-answer pairs, with an average video duration of 17 minutes. 2) **LVBench** Wang et al. (2024b) is an hours-long benchmark with an average length of 4101 seconds (68 minutes), 1549 question-answer pairs, and four multiple-choice options. 3) **HourVideo** Chandrasegaran et al. (2024), we use its dev set, including 50 videos with an average duration of 47.2 minutes, comprising 1182 high-quality, five-way multiple-choice questions. 4) **Egoschema** Mangalam et al. (2023) is a popular benchmark derived from Ego4D Grauman et al. (2022). It consists of 5-way multiple-choice questions on videos, which are 180 seconds long. We run ablations on the subset of 500 labelled examples.

**Implementation Details.** We evaluate the popular video-based MLLMs as our baselines, namely, Qwen2VL-7B Wang et al. (2024a) and Qwen2.5VL-7B Bai et al. (2025) on all datasets under the multiple-choice QA setting. For each video of various benchmarks, we densely sample frames at 1

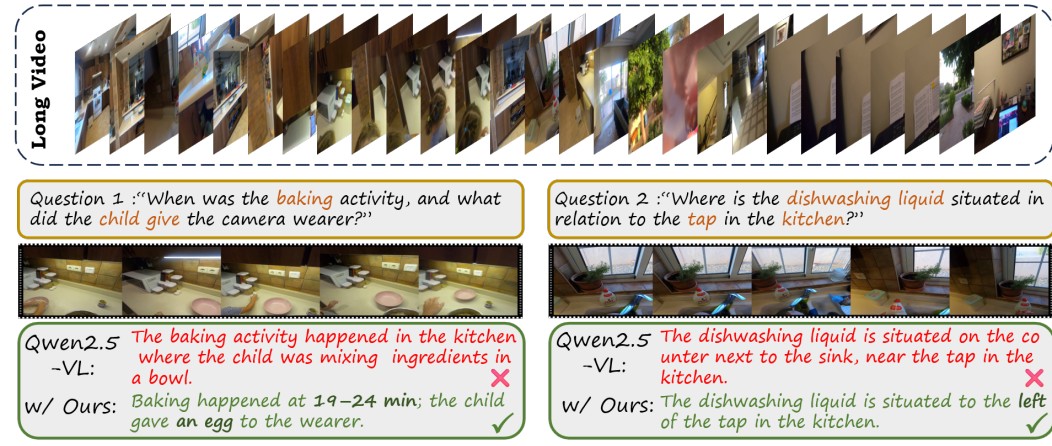

Figure 3: Qualitative examples from the HourVideo dataset comparing our model (answers in green) with the Qwen2.5-VL baseline (predictions in red), where displayed frames are hand-selected to illustrate key events relevant to the query.

frame per second (fps), collecting up to 1024 frames per video. From these, we use CLIP with the ViT-G backbone Radford et al. (2021) for semantic retrieval. To build our model, we leverage five foundation models: DINOV2 Oquab et al. (2023) and RAFT Teed & Deng (2020) for computing spatio-temporal differences, Grounding-DINO Liu et al. (2023) for object detection, Tarsier2-7B Yuan et al. (2025b) to capture clip-level dynamic narratives, and Qwen3-8B Yang et al. (2025) is used as the agent for CoT reasoning. Experiments are conducted on an NVIDIA A800 GPU.

### 4.2 BENCHMARK RESULTS

We report results using Qwen2-VL Bai et al. (2025) and Qwen2.5-VL Wang et al. (2024a) as backbones to show the effectiveness of Video-EM. Additionally, we use LLaVA-OV Li et al. (2025) and LLaVA-Video Zhang et al. (2024b) as the backbone to validate its versatility across mainstream models (see Appendix). The best result is in bold and the second-best is underlined.

**Comparison with State-of-the-Art Methods.** We evaluate the effectiveness of Video-EM across four representative long-form video benchmarks: Video-MME, LVBench, HourVideo, and Egoschema. As shown in Table 1, compared with keyframe sampling methods on Video-MME, such as AKS Tang et al. (2025) and Q-Frame Zhang et al. (2025c), Video-EM consistently outperforms existing training-free approaches and achieves highly competitive results among open-source models on all four benchmarks. Notably, on challenging long-video datasets such as LVBench and HourVideo, Video-EM improves accuracy up to 9% compared to baselines, demonstrating its effectiveness for long-form video reasoning. Furthermore, on the Egoschema benchmark (Table 2), our framework shows notable effectiveness compared to the baseline, improving performance to 65.6% and 64.4%, respectively, while reducing the number of frames used from 16 to 9. These results highlight the strong generalization capabilities of Video-EM, significantly outperforming existing open-source state-of-the-art methods and underscoring its adaptability to complex, egocentric video scenarios. Overall, these results demonstrate that Video-EM, as a training-free framework, effectively mitigates the limitations of previous keyframe-based methods, and is both widely applicable and robust across a range of long-video understanding tasks.

**Improvement Results on Off-the-Shelf VLMs.** As shown in Table 1, our proposed Video-EM framework consistently enhances the performance of off-the-shelf Vision Language Models (VLMs) without requiring training. When applied to mainstream models like Qwen2-VL Bai et al. (2025) and Qwen2.5-VL Wang et al. (2024a) across four long-form video benchmarks, we observe significant gains. For instance, on the Video-MME benchmark, Video-EM improves the accuracy of Qwen2.5-VL from 59.2% to 62.0% using only 28 frames. This demonstrates the effectiveness of our strategy, with qualitative examples provided in Figure 3.

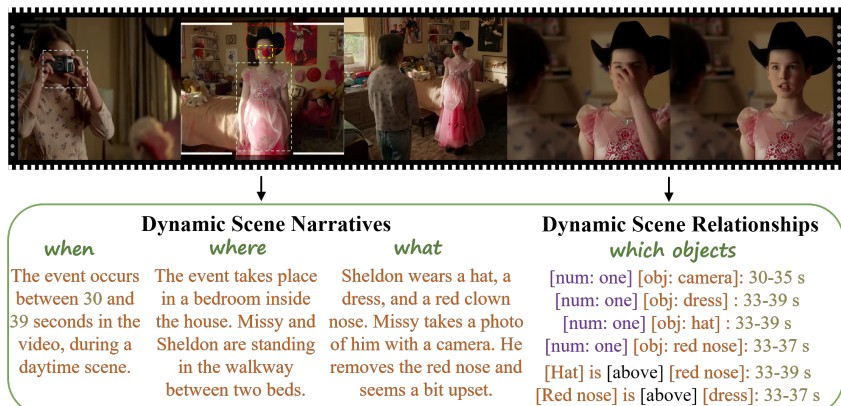

**Figure 4:** Qualitative examples of our Episodic Memory, composed of Dynamic Scene Narratives and Dynamic Scene Relationships.

**Table 3:** *Left*: Ablations of different Video-EM components. EMR refers to Episodic Memory Representation. In EMR, AEE refers to Adaptive Event Expansion, DSN to Dynamic Scene Narratives, and DSR to Dynamic Scene Relationships. *Right*: The adaptive event expansion module shows performance across different threshold settings.

| ID | Settings | Frames | Acc (%) |
|----|----------|--------|---------|
| 1 | **Full** | **Avg 9** | **64.4** |
| 2 | w/o EMR | Avg 8 | 59.0 |
| 3 | w/o AEE | Avg 8 | 63.2 |
| 4 | w/o DSN | Avg 9 | 59.0 |
| 5 | w/o DSR | Avg 9 | 62.8 |
| 6 | w/o CoT | Avg 41 | 62.8 |

| Metric | Threshold | Acc (%) |
|--------|-----------|---------|
| $D_{spatial}$ | 1.0 | 63.6 |
| | **2.0** | **64.4** |
| | 3.0 | 63.2 |
| $D_{flow}$ | 1.0 | 63.4 |
| | **2.0** | **64.4** |
| | 3.0 | 63.6 |

## 4.3 ABLATION STUDIES

We ablate on Egoschema using Qwen2.5-VL as the backbone to assess the contributions of Video-EM model variants.

**Ablation on Core Components.** We evaluate the individual contributions of Video-EM's core components: Episodic Memory Representation (EMR), Adaptive Event Expansion (AEE), Dynamic Scene Narratives (DSN), Dynamic Scene Relationships (DSR), and the Chain-of-Thought (CoT) reasoning module. As shown on the left side of Table 3, removing EMR causes performance to drop from 64.4% to 59.0% (ID = 1 *vs.* ID = 2), underscoring the vital role of structured episodic memory in grounding high-level video reasoning. Our EMR is illustrated in Figure 4. Within EMR, eliminating the AEE module (ID = 3) results in reliance solely on similarity-based keyframe retrieval. This neglects surrounding contextual information and often captures only partial events, leading to suboptimal coverage and a large performance drop. Furthermore, removing the CoT module (ID = 6) dramatically increases the number of selected frames, from 9 to 41, while the accuracy decreases from 64.4% to 62.8%. This suggests that overloading the model with redundant visual inputs can degrade reasoning efficiency and accuracy. The result highlights that selecting a minimal yet semantically rich subset of frames is more effective than processing redundant visual content.

**Ablation on the Spatio-temporal Difference Threshold.** Furthermore, we evaluate the robustness of Video-EM with respect to variations in the spatio-temporal threshold used in the Adaptive Event Expansion module. As shown in Table 3 (right), our method demonstrates stable performance across a range of thresholds. By appropriately tuning this hyperparameter, Video-EM achieves optimal performance, indicating its resilience to changes in event boundary sensitivity.

**Effect of Varying Number of Frames.** To assess the role of temporal granularity, we systematically vary the frame budget and evaluate performance on two long-video benchmarks—Video-MME and LVBench. As reported in Table 5, accuracy improves monotonically within our evaluated range as more frames are provided, underscoring the value of denser temporal sampling. The additional

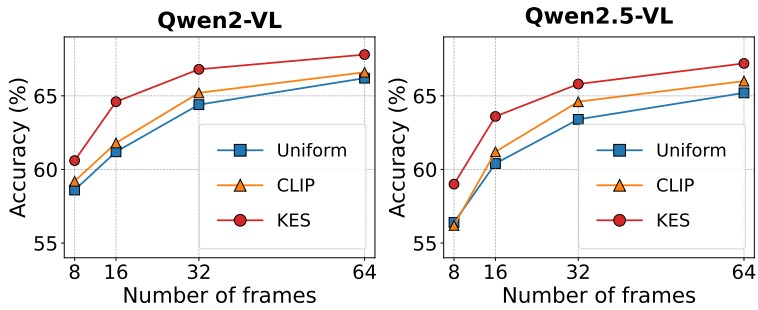

Figure 5: Ablation on the number of frames under different sampling strategies.

Table 4: Hyperparameter Analysis in the CoT Module.

| Settings | Value | Acc (%) | Settings | Value | Acc (%) |
|---|---|---|---|---|---|
| $D_{max}$ | 2 | 63.6 | $\tau$ | 0.8 | 63.8 |
| | **3** | **64.4** | | 0.7 | 63.4 |
| | 4 | 63.2 | | **0.6** | **64.4** |
| | 5 | 63.2 | | 0.5 | 63.1 |

frames supply finer motion cues and more complete coverage of event boundaries, which in turn yields more reliable reasoning over long-form narratives.

**Ablation on the CoT Module.** We further investigate the influence of key hyperparameters in the CoT module—namely, the maximum reasoning depth ($D_{max}$) and the confidence threshold ($\tau$), which govern the iterative selection and refinement of episodic memories. As summarized in Table 4, Video-EM maintains consistently strong accuracy across a wide spectrum of parameter values, demonstrating that its performance is highly robust and largely insensitive to hyperparameter tuning. This resilience underscores the practicality of our framework, as it can be readily applied without the need for extensive parameter optimization.

Table 5: Evaluation of Video-EM performance with varying numbers of input frames.

| Dataset | Max_num_frames | Acc (%) |
|---|---|---|
| Video-MME | Avg 7 | 56.7 |
| | Avg 14 | 59.4 |
| | Avg 28 | **62.0** |
| LVBench | Avg 7 | 37.3 |
| | Avg 13 | 40.6 |
| | Avg 27 | **45.7** |

**Effect of Keyframe Selection Strategies.** We assess the performance of three keyframe selection strategies: Uniform sampling, CLIP-based similarity retrieval, and our Adaptive Expansion Strategy (KES) on the Egoschema benchmark (Figure 5). Compared to uniform sampling and the naive CLIP-based approach, KES decomposes each query into multi-granular semantic cues to retrieve more contextually relevant frames, resulting in a substantial accuracy gain.

## 5 CONCLUSION

In this paper, we present Video-EM, a training-free framework designed to advance long-form video understanding in Video-LLMs. Unlike conventional approaches that treat keyframes as isolated snapshots, Video-EM reformulates question answering by organizing them into temporally ordered events, encoding these events as episodic memories, and applying a chain-of-thought strategy to extract a minimal yet maximally informative subset. This design enables more sophisticated spatio-temporal reasoning while maintaining efficiency. As a plug-and-play module, Video-EM integrates seamlessly with diverse Video-LLM backbones, requiring no costly retraining or architectural modifications. Comprehensive experiments on four mainstream long-video benchmarks highlight substantial performance gains, demonstrating both the effectiveness and robustness of our approach.

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

| Model | LLM Size | Frames | LongVideoBench$_{val}$ | MLVU$_{val}$ |
|---|---|---|---|---|
| *Video Duration* | | | 12 min | 12 min |
| LLaVA-NeXT | 7B | 64 | - | 42.3 |
| PLLaVA | 7B | 64 | 40.2 | 44.6 |
| VideoChat2 | 7B | 64 | 39.3 | 44.5 |
| ShareGPT4Video | 8B | 32 | 41.8 | 46.4 |
| Video-LLaMA2 | 7B | 32 | - | 48.5 |
| LongVU | 7B | 1 fps | 65.4 | - |
| QWen2-VL | 7B | 16 | 51.2 | 41.4 |
| QWen2-VL | 7B | 32 | 52.6 | 45.8 |
| **+ Video-EM** | 7B | Avg 24 | 55.4 | 49.4 |
| QWen2.5-VL | 7B | 16 | 54.3 | 43.8 |
| QWen2.5-VL | 7B | 32 | 56.8 | 48.2 |
| **+ Video-EM** | 7B | Avg 24 | 59.6 | 51.4 |
| LLaVA-OV | 7B | 16 | 50.8 | 44.6 |
| LLaVA-OV | 7B | 32 | 51.2 | 51.8 |
| **+ Video-EM** | 7B | Avg 24 | 53.6 | 54.2 |
| LLaVA-Video | 7B | 16 | 51.5 | 48.4 |
| LLaVA-Video | 7B | 32 | 52.1 | 51.8 |
| **+ Video-EM** | 7B | Avg 24 | 53.9 | 54.4 |

Table 6: Quantitative comparisons on the LongVideoBench and MLVU benchmarks.

## A APPENDIX

In this appendix, we present a comprehensive overview that includes the limitations of our work, additional quantitative results, extended ablation studies on the Video-EM framework, further implementation details, and supplementary qualitative analyses.

## B LIMITATIONS

Like other LLM-based video reasoning systems—including those relying on dense or CLIP-based sampling—our method is limited by the accuracy of captioners and object detectors in capturing the dynamic content of sampled events. However, Video-EM's modular, episodic memory architecture enables easy integration of improved captioning models, detection modules, and more powerful LLMs as they emerge.

While Video-EM is entirely training-free, it does involve a small number of hyperparameters. In the Experiment Section in the main paper, we analyze their effects and show that Video-EM consistently outperforms both uniform and CLIP-based baselines, regardless of the specific choices for maximum depth or branch width. This robustness highlights the effectiveness of our episodic memory framework, even under suboptimal configurations, for structured and scalable long-form video understanding.

## C ADDITIONAL QUANTITATIVE RESULTS

### C.1 ADDITIONAL EVALUATION BENCHMARKS.

As shown in Table 6, we comprehensively evaluate our Video-EM framework using four widely adopted and representative vision-language backbones: QWen2-VL, QWen2.5-VL, LLaVA-OV, and LLaVA-Video. These models span diverse architectural designs and capabilities, providing a robust foundation for assessing the generalizability and effectiveness of Video-EM across different settings.

---

**Algorithm 1** The pseudo code of Video-EM.

---

**Input**: Video $V$, Question $q$, Options $\{o_0, o_1, o_2, o_3\}$
**Parameter**: $K, \tau, D_{max}$
     Episodic Memory $EM_i \in EM$
**Output** : Answer $\in \{o_0, o_1, o_2, o_3\}$

---

**Key Event Selection (KES):**

**1:**     $Q =$ Decompose $(q)$           ▷ multi-grained query set
**2:**     $V^* = \text{Top}K(\text{CLIP}(Q, V))$           ▷ key frames
**3:**     $\mathbf{E}^* = \text{Event\_Segmentation}(V^*)$           ▷ key events

**Episodic Memory Representation (EMR):**

**4:**     $\mathbf{E} = \text{Event\_Expansion}(\mathbf{E}^*)$           ▷ expanded events
**5:**     **Function** Epi\_Mem\_Construct $(\mathbf{E}_i, prompt)$
**6:**       $\mathbf{N}_i^{scene} \leftarrow \text{Dynamic\_Scene\_Narratives}(E_i, prompt)$
**7:**       $\mathbf{G}_i^{scene} \leftarrow \text{Dynamic\_Scene\_Relationships}(E_i)$
**8:**       return $EM_i \leftarrow (\mathbf{G}_i^{scene}, \mathbf{N}_i^{scene})$           ▷ episodic memory
**9:**     **for**   $E_i \in \mathbf{E}$ **do**
**10:**       $EM$.append(Epi\_Mem\_Construct $(E_i, prompt)$)

**Chain-of-Thought Thinking (CoT):**

**11:**     **Function** CoT\_Recurse $(EM, E, q, depth)$
**12:**       (reasoning, conf) $\leftarrow$ Predict\_CoT$(EM, q, depth)$
**13:**       **if** conf $\geq \tau$ **or** $depth \geq D_{max}$ **then return** $EM, E$
**14:**       $[E^L, E^R] \leftarrow \text{Cluster}(E)$
**15:**       $EM^L = \text{Epi\_Mem\_Construct}(E^L, prompt)$
**16:**       $EM^R = \text{Epi\_Mem\_Construct}(E^R, prompt)$
**17:**       **if** CoT\_Recurse $(EM^L, E^L, q, depth + 1)$
        **is valid then return** $EM^L, E^L$
**18:**       **else return** CoT\_Recurse $(EM^R, E^R, q, depth + 1)$
**19:**     **for**   $EM_i \in EM$ **do**
**20:**       Selected\_EM.append(CoT\_Recurse $(EM_i, E_i, q, 1)$)
**21:**     Answer $\leftarrow$ VideoLLM(Selected\_EM, $q$, $(o_0, o_1, o_2, o_3)$)

---

**MLVU Results.** On the MLVU validation benchmark featuring 12-minute videos, our Video-EM module delivers consistent and substantial enhancements across all four mainstream backbones: QWen2-VL, QWen2.5-VL, LLaVA-OV, and LLaVA-Video.

Notably, these improvements are achieved with an adaptive average of just 24 frames, which is fewer than the 32-frame baselines while yielding superior performance. Specifically, QWen2.5-VL + Video-EM attains 51.4% accuracy, marking a 3.2 percentage point gain over its 32-frame baseline (48.2%) and a 7.6 point improvement over the 16-frame version (43.8%).

LLaVA-Video + Video-EM achieves the highest score of 54.4%, outperforming its 32-frame baseline by 2.6 points (51.8%) and the 16-frame baseline by 6.0 points (48.4%).

Likewise, QWen2-VL + Video-EM boosts performance to 49.4%, representing a 3.6 point increase from the 32-frame baseline (45.8%) and an 8.0 point uplift from the 16-frame baseline (41.4%).

These advancements highlight Video-EM's ability to enhance overall video understanding by overcoming the limitations of fixed-frame inputs through efficient and adaptive spatio-temporal modeling, enabling richer contextual comprehension for accurate long-form video understanding.

**LongVideoBench Results.** The Video-EM module similarly yields significant performance boosts on the LongVideoBench validation benchmark for 12-minute videos: QWen2.5-VL + Video-EM reaches 59.6%, exceeding its 32-frame baseline by 2.8 points (56.8%) and the 16-frame baseline by 5.3 points (54.3%).

QWen2-VL + Video-EM achieves 55.4%, improving by 2.8 points over the 32-frame version (52.6%) and 4.2 points over the 16-frame version (51.2%).

LLaVA-OV + Video-EM records 53.6%, surpassing its 32-frame baseline by 2.4 points (51.2%) and the 16-frame baseline by 2.8 points (50.8%).

LLaVA-Video + Video-EM records 53.9%, surpassing its 32-frame baseline by 1.8 points (52.1%) and the 16-frame baseline by 2.4 points (51.5%).

Positioned among the top performers in the comparative table, all utilizing 7B LLM sizes, these Video-EM-enhanced models outshine competitors such as Video-LLaMA2 (48.5% on MLVU) and ShareGPT4Video (41.8% on LongVideoBench), despite employing fewer or comparable frames (average 24).

## C.2 Additional Dataset Details

**MLVU Zhou et al. (2025):** Multi-task Long Video Understanding Benchmark is a new dataset designed to evaluate Long Video Understanding performance. It addresses the limitations of existing benchmarks by offering longer video durations, diverse video genres (such as movies, surveillance footage, and cartoons), and a range of evaluation tasks. The benchmark includes 2593 tasks across 9 categories, with an average video duration of 12 minutes, providing a comprehensive assessment of MLLMs' capabilities in understanding long videos. This allows for a more comprehensive assessment of MLLMs' capabilities in understanding long videos.

**LongVideoBench Wu et al. (2024):** It is a recent benchmark designed to evaluate long-term video-language understanding for MLLMs. It consists of 3763 web-collected videos of varying lengths, up to one hour, with subtitles, covering a wide range of themes. The dataset is tailored to assess models' ability to process and reason over detailed multimodal information from long video inputs. It includes 6678 human-annotated multiple-choice questions across 17 fine-grained categories, making it one of the most comprehensive benchmarks for long-form video understanding. In this paper, we focus on the validation set without subtitles, denoted as $\text{LVB}_{val}$, which contains 1337 question-answer pairs and has an average video length of 12 minutes.

## D The Pseudo Code

This section formalizes our framework and presents the pseudo-code used to produce query-aware, temporally grounded episodic memories for long-form videos. As illustrated in Alg. 1, we first summarize the notation, then provide the core algorithm and implementation notes.

## E CoT

As illustrated in Figure 6, we present the complete trajectory of CoT-based reasoning, highlighting how our framework progressively builds an understanding of long-form videos. At each step, the agent adaptively evaluates segment-level confidence scores, selectively retaining the most reliable excerpts, decomposing complex events into finer temporal units, and capturing subtle contextual and relational cues. This iterative process yields rich, temporally grounded representations that provide Video-LLMs with a more nuanced foundation for downstream reasoning and question answering.

## F Prompt

We present detailed prompt designs to illustrate our method.

    A. is the prompt we used for Query Decomposition.

    B. is the prompt we used for Question Answering.

    C. is the prompt we used for Dynamic Scene Narrative.

    D. is the prompt we used for the CoT Thinking Prompt.

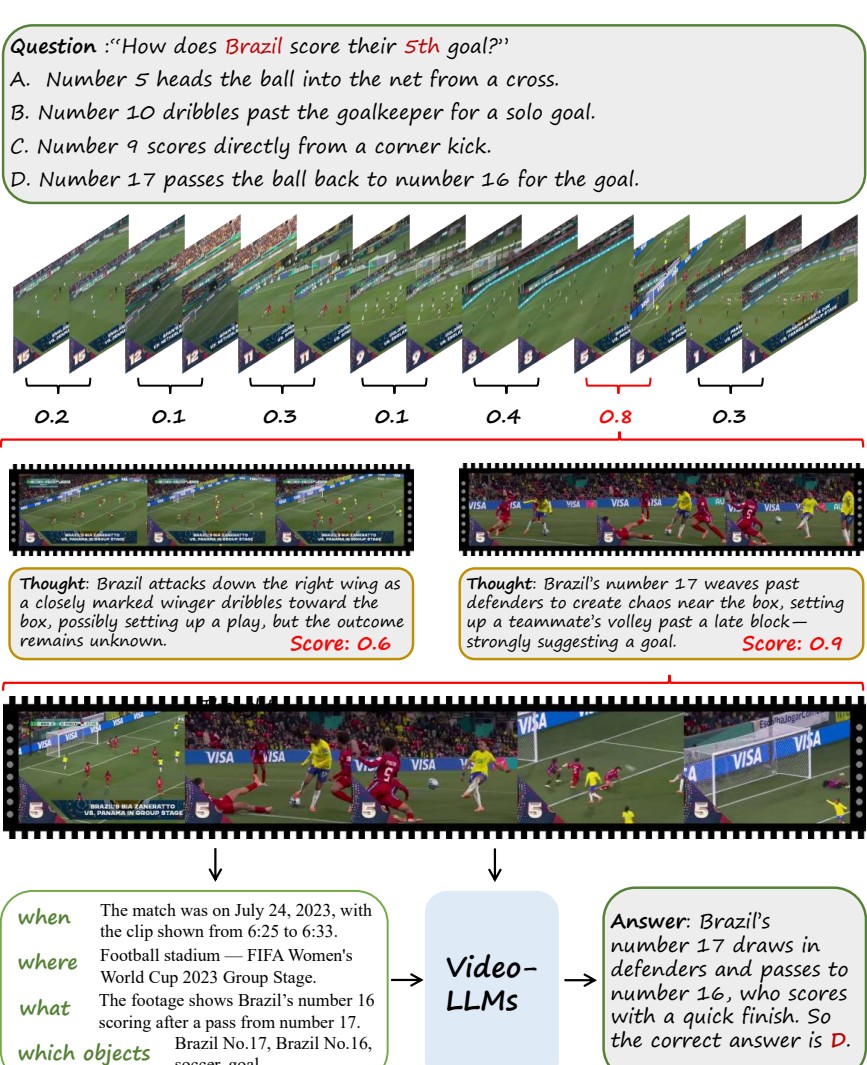

Figure 6: CoT Thinking Trajectory Example. Our agent typically selects video segments with higher confidence scores for exploration. When finer-grained analysis is needed, the CoT framework adaptively decomposes the selected event into temporally refined episodic memory representations (when, where, what, and which objects), allowing the model to capture subtle transitions and nuanced contextual cues.

## G  DECLARATION OF LLM USAGE

In preparing this manuscript, we made use of Large Language Models (LLMs), such as ChatGPT and similar systems. Their role was limited to language-oriented tasks, including grammar refinement, spelling correction, and word choice optimization, thereby improving the clarity and readability of the paper. Importantly, all core scientific contributions—including the development of ideas, dataset construction, benchmark design, analyses, and conclusions—were entirely conceived, executed, and validated by the authors.

**A. Prompt Template for Query Decomposition**

Analyze the following input question:
Question: `<Question>`
Options: `<Options>`

**Process**: Key Object/Scene Identification

- Extract key objects/scenes from the input query
- Format: Key Objects: `obj1, obj2, obj3`
- Format: Key Scenes: `Scene1, Scene2, Scene3`

**Output Rules**

1. One line each for Key Objects and Scenes, starting with exact prefixes
2. Separate items with a comma
3. Never use markdown or natural language explanations
4. If you cannot identify any key objects or cue objects from the query provided, please output null

**Below is an example of the procedure:**

Question: For "When does the person in red clothes appear with the dog at the fence?"
Response:
  Key Objects: `person, dog, red clothes`
  Cue Scenes: `grassy_area, fence`

**Format your response EXACTLY like this in two lines:**

  Key Objects: `[object1, object2, object3]`
  Key Scenes: `[Scene1, Scene2, Scene3]`

---

**B. Prompt Template for Question Answering**

Select the best answer to the following multiple-choice question based on the video.

Event 1:
  `[[<image> <image>  ··· <image> ],`
  `[EM Representation:` *when, where, what, which objects*`]]`
···

Event $N$:
  `[[<image> <image>  ··· <image> ],`
  `[EM Representation:` *when, where, what, which objects*`]]`

Question: `<Question>`

Options: `<Options>`

Answer with the option's letter from the given choices directly.
Your response format should be strictly an upper case letter A, B, C, D or E.

**C. Prompt Template for Dynamic Scene Narrative**

You are a video-captioning agent whose goal is to write a single-event memory for each clip. Describe only what is visible or audibly obvious; do not infer motives or facts outside the frame.

**Output Rules**

1. Output exactly three lines starting with when:, where:, and what:, in this order, each beginning with the specified label in bold, followed by a colon and a single sentence.

2. Use the present tense, no more than 30 words per line, and include concrete details that make the event uniquely retrievable later.

**When:** [State the explicit temporal cue visible or audible (time-of-day lighting, season indicators, on-screen date stamp), and specific times in the video.]

**What:** [Describe the primary action, key actors or objects involved, and any immediately resulting outcome. Use specific verbs; mention counts, colors, or notable props when clear.]

**Where:** [State the precise setting: indoor/outdoor, venue type (e.g., "city street", "kitchen"), and any landmark, sign, or environmental cue visible.]

**Important Notes**

1. If any element cannot be determined from the clip alone, write "Unknown" for that line.

2. Avoid generic words like "someone", "place", "day" when more specific details are observable.

3. Do not add extra lines, punctuation, or commentary.

**Below are the examples of the procedure:**

**Example 1:**

**When:** [Morning, sunlight filtering through eastern window, spring daffodils on sill. The main activity occurs between [33 s-52 s] in the video.]

**Where:** [Homey farmhouse kitchen with wooden beams, floral curtains, and cast-iron stove.]

**What:** [Elderly woman in polka-dot apron sprinkles powdered sugar onto a stack of steaming chocolate crêpes.]

**Example 2:**

**When:** [Late afternoon on a cloudy autumn day, fallen leaves scattered. The main activity occurs between [2min46s-3min12s] in the video.]

**Where:** [Open plaza in front of a stone-arched European town hall.]

**What:** [Two children in red raincoats chase a yellow balloon across a puddled cobblestone square.]

## D. Chain-of-thought Thinking Prompt

**Role and Goal**

You are an expert AI agent specializing in cognitive systems. Your primary function is to act as a relevance evaluation module. Your goal is to analyze a user's Query and an Episodic Memory representation of a past event. You must determine how relevant the memory is to answering the query and express this as a confidence score. You must also provide a detailed, step-by-step "chain-of-thought" reasoning for your evaluation.

**Input Specification**

You will receive two pieces of information in a structured format:
• Query: A natural language string representing the user's question.
• Episodic Memory: A structured object representing a single past event, containing the following fields:
• When: (String/Timestamp) The time the event occurred (e.g., "2023-10-27 09:00:00", "yesterday morning").
• Where: (String) The location where the event took place.
• What: (String) A description of the core action or event that occurred.
• which_objects: (List of Strings) A list of key objects involved in or present during the event.

**Task & Output Specification**

Your task is to generate a JSON object containing two key-value pairs:

confidence_score: A floating-point number representing the relevance of the Episodic Memory to the Query. This score must be in the range $\in [0, 1]$, where 1 indicates perfect relevance and 0 indicates no relevance.

chain_of_thought: A detailed, step-by-step explanation of your reasoning process. This should follow the "Chain-of-Thought" methodology outlined below.

**Chain-of-Thought Methodology**

You must strictly follow these five steps to arrive at your conclusion. Articulate each step clearly in your output.

Step 1: Deconstruct the Query.

Analyze the user's Query to extract its core semantic components. Identify the key intent.
Break the query down into its own implicit what (action/event), when (timeframe), where (location), and which_objects components. If a component is not mentioned in the query, state that it is "unspecified" or "broad."

Step 2: Component-wise Matching and Scoring.

Compare the components extracted from the Query (from Step 1) with the corresponding fields in the provided Episodic Memory.
For each of the four components (what, when, where, which_objects), perform a semantic comparison and assign a partial relevance score (S) from $[0, 1]$:
S_what: How semantically similar is the event in the memory to the action in the query? (e.g., "made a meal" is highly relevant to "cooked dinner").
S_where: How well does the memory's location match the query's location? (e.g., "office" matches "work"; "kitchen" is a subset of "home").
S_when: Does the memory's timestamp fall within the timeframe specified by the query? (e.g., "Tuesday at 3 PM" falls within "this week").
S_objects: What is the degree of overlap between the objects in the memory and the objects in the query? Consider direct matches and semantic relationships.

Step 3: Determine Component Weights based on Query Focus.

Analyze the Query again to determine which components are most critical for answering it.
For example, for the query "Where did I put my keys?", the where and objects components are most important, so they should receive higher weights. For "What was I doing yesterday?", what and when are most important.

Step 4: Calculate the Final Confidence Score.

Calculate the final confidence score as the weighted average of the partial scores from Step 2, using the weights from Step 3.
Use the following formula:
Confidence_Score = S_what + S_where + S_when + S_objects
Round the final score to 3 decimal places.

Step 5: Justify the Final Score.

Provide a brief, concluding summary that explains why the final confidence_score is what it is. Synthesize the findings from the previous steps, highlighting the strongest points of relevance and the key mismatches that lowered the score.

