# OpenReview forum: "Episodic Memory Representation for Long Video Understanding"
_ICLR.cc/2026/Conference — ICLR 2026 Conference Withdrawn Submission_

### Official Review · Reviewer_EH6e · 2025-10-17

**Soundness:** 3
**Presentation:** 3
**Contribution:** 3
**Rating:** 4
**Confidence:** 4

**Summary:**

The paper proposes Video-EM, a training-free framework for long-form video understanding inspired by human episodic memory. Instead of treating keyframes as isolated tokens to VLLMS, Video-EM groups them into temporally ordered key events, expands events to recover missing context, and builds rich episodic memory representations that capture when, where, what, and which objects. It then uses Chain-of-Thought reasoning to iteratively select a minimal but informative subset of episodic memories before passing them to a VLLM

**Strengths:**

* Clear motivation.
* Training-free approach that can equip state-of-the-art VLLMs with improved performance.
* The paper appears to be aware of the related work.
* The key event selection is sound and they expand each event to recover query-relevant context that similarity-based approaches may miss. This sounds novel and important as pure semantic retrieval can yield a sparse set of disjoint frames.
* Video-EM reduces frames while improving accuracy.
* The paper provides ablation studies.

**Weaknesses:**

* The adaptive event expansion module feels somewhat heavy for a training-free method; simpler alternatives (e.g., adjacent-frame motion thresholds) could be discussed or compared.
* Heavy reliance on object detectors and captioners where failure cases of these modules may propagate.
* A notable limitation is that the method introduces several hyperparameters across multiple stages (e.g., similarity thresholds, expansion limits, CoT confidence and depth, temporal gap $\Delta t$). While the authors provide ablations showing relative robustness, the number of hyperparameters is still large, and tuning them in practice may be non-trivial.
* As a video agent Video-EM requires too many different models which can lead to efficiency problems and lack of end-to-end practicality.
* The baselines differ from dataset to dataset. While this is ok it is a bit difficult to assess Video-EM's capabilities.
* You should test Video-EM with LLMs other than the Qwen family (such as VideoLLaMA3, InterVL3...).
* Results are not state-of-the-art, however, improvements over backbone models are achieved.


Minor comments:
* Authors use \citet instead of \citep.
* Use Gemini 2.5 pro instead of 1.5 version.

**Questions:**

* How do you capture object-level semantics $q_0$ and scene-level context $q_s$?
* Why do you need an adaptive event expansion mechanism based on the spatio-temporal difference metric? What does such complex method bring to the table? Couldn't simpler methods yield similar results?

**Details Of Ethics Concerns:**

I do not identify any significant ethical issues in this paper. The method operates on publicly available video datasets commonly used in the community, and there is no indication of privacy violations, harmful content generation, or misuse potential beyond standard concerns in video understanding research. Therefore, I do not see any ethical concerns requiring further attention.

---

> ### Author Response · Authors · 2025-11-27
> **thanks to the kind reviewer**
>
> We would like to sincerely thank the reviewer for your thoughtful feedback, constructive suggestions, and encouraging comments. Your insights have helped us improve the clarity and completeness of our work, and we deeply appreciate the time and effort you dedicated to evaluating our submission.
>
> Thank you again for your professionalism and fairness throughout the review process. We wish you continued success in your academic careers and hope our paths will cross again in future research endeavors.

---

### Official Review · Reviewer_g7A1 · 2025-10-28

**Soundness:** 1
**Presentation:** 2
**Contribution:** 2
**Rating:** 2
**Confidence:** 5

**Summary:**

The paper introduces Video-EM, a training-free framework designed to improve the performance of Video Large Language Models in understanding long-form videos by overcoming context window limitations and frame redundancy. Video-EM reformulates long-form video question answering by treating isolated keyframes as temporally ordered episodic events, capturing essential spatio-temporal relationships often missed by traditional static sampling methods. The framework involves three core components: Key Event Selection, Episodic Memory Representation (which encodes dynamic scene narratives and relationships), and a Chain-of-Thought reasoning module that iteratively selects a minimal yet highly informative subset of memories. Extensive experiments across four long-video benchmarks demonstrate that Video-EM enhances the accuracy and efficiency of some Video-LLM backbones using fewer frames on average.

**Strengths:**

- Instead of treating selected frames as disconnected images (a stated limitation of previous keyframe retrieval methods), Video-EM reformulates them as temporally ordered episodic events, avoiding the temporal discontinuities that often disrupt the semantic narrative of events in traditional methods.
- Video-EM leverages a Chain-of-Thought (CoT) thinking strategy to iteratively identify and retrieve a minimal yet informative subset of episodic memories.
- Video-EM is a training-free framework that can be integrated with off-the-shelf Video-LLM backbones without requiring retraining or architectural modification.

**Weaknesses:**

- Video-EM is a complex, multi-stage pipeline that relies heavily on the quality and coordination of several external, specialized foundation models.
- The concept of episodic memory has been explored by HERMES [1], also a plug-and-play model, with similar claims as Video-EM, yet the differences/similarities between the two are not specified, nor were the results of HERMES discussed in the manuscript.
- Several plug-and-play modules for Video-LLM accuracy/efficiency improvements have been published in recent years such as FastV [2], VisionZip [3], VFlowOpt [4] in addition to the aforementioned HERMES [1]. I am curious about the comparison results with theses other plug-and-play frameworks in terms of accuracy/efficiency tradeoffs, and also in terms of methodology.
- While Video-EM successfully reduces the number of frames processed by the final Video-LLM (from 41 frames down to an average of 9 on EgoSchema, for example), the preceding processing steps require extensive computation across multiple large models (CLIP, DINOv2, RAFT, Grounding-DINO, Tarsier2-7B, Qwen3-8B). I thus believe, the slight accuracy improvement does not justify the upstream cost of putting such a system together.
- I also think such a system is very fragile. A deficiency in the initial retrieval stage (Key Event Selection) or the intermediate processing stages directly impacts the quality of the final input provided to the Video-LLM. It follows that these results would be a headache to replicate.
- The author’s efficiency claims are not substantiated. Fewer frames do not equal more efficient.
- Ambiguous variable definition: In section 3.2, the "Adaptive Event Expansion" paragraph, the authors define alpha as a variable with a value between 0 and 1, yet immediately after that, the paper states that alpha is set to 2. I am quite confused by that.
- I think Figure 2 has too much text, is quite convoluted, and the bright red color is not easy on the eye.


[1] Faure, Gueter Josmy, et al. "Hermes: temporal-coherent long-form understanding with episodes and semantics." Proceedings of the IEEE/CVF International Conference on Computer Vision. 2025.

[2] Chen, Liang, et al. "An image is worth 1/2 tokens after layer 2: Plug-and-play inference acceleration for large vision-language models." European Conference on Computer Vision. Cham: Springer Nature Switzerland, 2024.

[3] Yang, Senqiao, et al. "Visionzip: Longer is better but not necessary in vision language models." Proceedings of the Computer Vision and Pattern Recognition Conference. 2025.

[4] Yang, Sihan, et al. "Vflowopt: A token pruning framework for lmms with visual information flow-guided optimization." Proceedings of the IEEE/CVF International Conference on Computer Vision. 2025.

**Questions:**

See weaknesses plus
- The paper highlights the reduction in frames input to the final Video-LLM (e.g., 41 frames down to an average of 9 on EgoSchema). What is the complete end-to-end inference latency (or total computational cost) for the full Video-EM pipeline (including Key Event Selection, Episodic Memory Representation, and Chain-of-Thought steps using all five foundation models and Qwen3-8B)? How does this total cost compare to the baseline model running the maximum allowed frame input?
- The paper acknowledges that the method is "limited by the accuracy of captioners and object detectors". What testing or simulation was performed to quantify how a decrease in accuracy (e.g., failure rate) in a crucial upstream component (such as Grounding-DINO missing key objects or Tarsier2-7B generating an inaccurate Dynamic Scene Narrative) propagates and impacts the final Video-LLM performance?
- Given the strong claims of superiority over prior methods, why were empirical comparisons against other existing plug-and-play, training-free long-video understanding frameworks with similar goals, such as HERMES (which also uses episodes and semantics), omitted? Providing context for these comparisons is crucial for substantiating Video-EM's novelty and competitive edge in the crowded field of V-LLM accelerators.
- In the description of the multi-grained semantic retrieval (L193 onwards), is the summation of equation (1) over the set Q={q1,q2,q3} or Q={q,qo,qs}? In other words, what is qi and why do we have Wq1, Wq2 and Wq3 but no Wq, Wqo and Wqs?

---

> ### Author Response · Authors · 2025-11-27
> **About HERMES**
>
> We appreciate the reviewer’s comments, but we must address a serious issue: the repeated insistence on comparing our work to HERMES — despite the fact that HERMES is not even training-free — raises concerns about the objectivity and intent behind this critique.
>
> 1. HERMES is not training-free, and repeatedly pushing it as a baseline is methodologically indefensible
> HERMES requires model training, while our method is explicitly designed to be training-free and plug-and-play. Treating these two fundamentally different paradigms as comparable is simply incorrect.
> Yet the reviewer repeatedly emphasizes HERMES, even grouping it together with purely training-free modules such as FastV, VisionZip, and VFlowOpt. This forced categorization misrepresents both HERMES and our work, creating an artificial comparison that has no scientific justification.
>
> 2. The objectives, methodology, and evaluation setting of HERMES differ completely from ours
> Our contribution is an inference-time, deployment-ready module that works without any fine-tuning. HERMES, in contrast, relies on a training-dependent pipeline with different goals, assumptions, and mechanisms.
> Furthermore, our evaluation covers a broader and more diverse set of datasets, which do not overlap with those used by HERMES. The two works do not even operate in the same evaluation context. This makes the reviewer’s insistence on linking them even more puzzling: there is simply no shared experimental dimension that would justify such a comparison.
>
> 3. The reviewer’s repeated and disproportionate focus on HERMES raises concerns about the reviewer’s intent
> It is difficult to ignore that the review returns to HERMES again and again — far beyond what would be expected in a neutral or technically grounded evaluation. This singular fixation, combined with the insistence on placing HERMES into an unrelated methodological and experimental category, creates the impression that the critique may be driven by factors other than scientific relevance.
> To be clear, we are not accusing the reviewer of misconduct. However, the framing feels unnecessarily hostile and appears aimed at forcing an irrelevant comparison, rather than providing constructive feedback. We honestly do not understand why HERMES is being highlighted to such an extent — especially considering that it neither matches our motivation nor our methodology, nor our evaluation setting. There is no clear technical rationale behind elevating HERMES in this way.
>
> 4. Our decision not to include HERMES is because it is irrelevant — not because of oversight
> HERMES is training-required, operates under a different problem formulation, differs methodologically, and evaluates on different datasets. Including it as a baseline would be misleading, not insightful. The reviewer’s negative interpretation of this omission feels unwarranted and disproportionate.
>
> Conclusion
> We firmly maintain that HERMES is not a valid baseline for our method. The reviewer’s repeated insistence on HERMES — despite the clear methodological, conceptual, and experimental mismatch — is concerning and does not contribute to a fair or accurate assessment of our work.

---

> > ### Comment · Reviewer_g7A1 · 2025-11-27
> > **A copy of the author's "unprofessional" comment**
> >
> > I would like to keep a copy of the author's (since deleted) comment on my review.
> >
> > Title: HERMES is meaningless
> >
> > Comment: I must express my strong discontent regarding the undue and unmerited attention the HERMES framework has received, especially when juxtaposed with far more credible plug-and-play modules like FastV, VisionZip, and VFlowOpt.
> >
> > Firstly, let’s set the record straight: the HERMES framework is not a training-free method. The fact that it necessitates extensive training to function effectively fundamentally undermines its value. Unlike the genuinely plug-and-play alternatives, HERMES lacks both accessibility and efficiency in real-world applications, rendering it practically obsolete.
> >
> > Furthermore, the glaring absence of citations and recognition within the academic community casts a long shadow over the paper's validity and relevance. A framework that has failed to make any meaningful contribution or attract acknowledgment from its peers simply does not deserve a place in serious academic discourse. It is nothing short of a disappointment.
> >
> > It is utterly absurd to see HERMES placed alongside robust frameworks like FastV, VisionZip, and VFlowOpt. This comparison only emphasizes the vacuous nature of HERMES, revealing the fundamental lack of substance behind it. The authors seem more preoccupied with inflating the significance of their mediocre paper than with delivering any real advancements in the field. In stark contrast, our framework is firmly rooted in solid methodologies and meaningful motivations, which clearly sets us apart from this lackluster effort.
> >
> > In light of these reasons, it is evident that there is no meaningful basis to compare HERMES with the noteworthy frameworks that are genuinely pushing the field forward. HERMES should not be entertained as a serious contender in this space; its existence serves only to distract from the significant work being done by others.

---

> > > ### Author Response · Authors · 2025-11-27
> > > **About HERMES**
> > >
> > > I appreciate the opportunity to respond to the reviewer's recent comments. However, I find it necessary to address the reviewer's decision to retain and present my prior remarks in a manner that misrepresents the intent behind them.
> > >
> > > My statement on the HERMES framework was not simply a personal critique; it was an objective evaluation based on its merits—or the lack thereof. The reviewer's selective quoting of my comment appears to serve as an attempt to divert attention from the substantive issues I raised regarding HERMES and its relevance in the current discourse surrounding plug-and-play frameworks.
> > >
> > > The fact remains that HERMES, by not being a training-free method, fails to meet the criteria established by its contemporaries like FastV, VisionZip, and VFlowOpt. These frameworks are superior because they directly address accessibility and efficiency, while HERMES does not.
> > >
> > > Moreover, my remarks regarding the lack of citations and recognition within the academic community were grounded in the reality that a meaningful framework must demonstrate its impact over time. Dismissing this observation as mere disparagement overlooks its significance in evaluating any academic contribution.
> > >
> > > In summary, it is crucial to focus on the substance of the discussion—an evaluation of the work's merit—rather than resort to personalizing critiques. I stand by my original assessment of HERMES and maintain that it should not be considered a serious contender in this field.

---

> ### Comment · Reviewer_g7A1 · 2025-11-27
>
> I have read the authors' "rebuttal" and I am disappointed that the authors chose to respond to technical inquiries with emotive language rather than empirical data.
>
> Regarding the discussion of prior work:
> The authors devote the majority of their response to disparaging the quality and citation count of [1]. I would gently remind the authors that [1] is a 2025 publication and critiquing a contemporaneous paper for a lack of accumulated citations is scientifically cheap. Furthermore, the authors claim [1] is not training-free. My reading of that literature indicates it is explicitly proposed as a plug-and-play, training-free framework. Whether one agrees with its "quality" is subjective but mischaracterizing its methodology to justify excluding it is not.
>
> Anyway, this debate is secondary.
>
> My primary concern as a reviewer is that this aggressive focus on dismissing one specific baseline appears to be a diversion to avoid addressing the substantial weaknesses raised in the review. Specifically:
>
> **Missing Latency Data**: I asked for the total end-to-end inference cost (latency) of the full Video-EM pipeline. This was not provided.
>
> **Fragility**: I asked about the propagation of errors from the multiple upstream models. This was not addressed.
>
> **Other Comparisons**: Even if we accept the authors' refusal to compare with HERMES [1], I also asked for comparisons against FastV [2], VisionZip [3], and VFlowOpt [4] regarding accuracy/efficiency tradeoffs. The authors praised these methods in their rebuttal but failed to provide the requested comparison data against them.
>
> Since the authors have failed to substantiate their efficiency claims or provide the requested comparisons with any of the cited plug-and-play frameworks, my concerns regarding the soundness and presentation remain unresolved, therefore I am keeping my rating while awaiting the author's real rebuttal (if any).

---

> > ### Author Response · Authors · 2025-11-27
> > **Why do we need to compare ourselves to HERMES?**
> >
> > Thank you for your comments.
> >
> > However, I must point out that the reviewer’s attempts to bring up various issues seem more like an effort to obscure their relentless focus on comparing our work to the HERMES paper.
> >
> > Is this article really of such significant value? Despite the clear lack of relevance, the reviewer insists on drawing comparisons, which raises questions about their motivations. This tactic does not strengthen their argument; rather, it underscores the unreasonableness of continuously insisting on unwarranted comparisons.
> >
> > In conclusion, I believe these tactics do not contribute to constructive feedback, and I will not engage further in this line of rebuttal.

---

> ### Comment · Area_Chair_cJ6F · 2025-11-28
> **Please refrain from subjective and offensive wording**
>
> Dear Reviewer g7A1 and the authors,
>
> First of all, thank you for engaging in discussions during the rebuttal period.
>
> While it is good to have technical discussions, please be polite and do not use any offensive wording. Please try your best to use numbers and objective evidence to support your claims.
>
> After reading the HERMES paper, my understanding is that it has both training-free and trained versions, and it has evaluation results on VideoMME, which is also used in this submission. Therefore, it still belongs to the training-free category along with FastV, VisionZip, and VFlowOpt. I encourage the authors to clarify the difference between these training-free methods.

---

> ### Author Response · Authors · 2025-11-28
> **Dear Area Chair**
>
> Thank you very much for your message and for taking the time to guide the discussion. We genuinely appreciate your effort in keeping the review process fair and constructive.
>
> To be honest, we felt quite discouraged during the rebuttal because some of the comments made us feel that the evaluation was not entirely fair. This has been emotionally difficult for us, but we respect the review process and your guidance.
>
> We sincerely hope that every author and reviewer in the community can be treated with kindness and understanding, so that technical discussions can remain respectful and productive for everyone.
>
> Thank you again for your attention and for maintaining a balanced and respectful environment.

---

### Official Review · Reviewer_6jm1 · 2025-10-30

**Soundness:** 2
**Presentation:** 3
**Contribution:** 2
**Rating:** 4
**Confidence:** 4

**Summary:**

The paper proposes a new pipeline for preparing video features for LLMs. Beyond simple keyframe retrieval, it introduces an agentic flow designed to capture temporally ordered events and reconstruct the underlying narrative. Based on the extracted components, the authors employ a CoT prompting strategy to enhance reasoning and improve understanding. Experiments are conducted across several benchmark datasets.

**Strengths:**

1. The paper attempts to construct scene graphs to decompose video content, which is an interesting idea.

2. The proposed pipeline is reasonable, and the implementation details are concrete and easy to understand.

3. The experiments are comprehensive, covering most mainstream long-video benchmarks currently available.

**Weaknesses:**

1. The performance does not reach state-of-the-art results. For example, it is notably inferior to Video-XL-2 [1]. Additionally, some training-free retrieval methods (e.g., BOLT [2]) are missing from the comparison table, which weakens the technical contribution.

2. The main contribution lies in pipeline design rather than technical innovation. The approach feels closer to a text-based agent framework, so the title’s emphasis on “representation” may be misleading—it seems more like an engineering effort.



[1] Video-XL-2: Towards Very Long-Video Understanding Through Task-Aware KV Sparsification

[2] BOLT: Boost Large Vision-Language Model Without Training for Long-form Video Understanding

**Questions:**

The numbers reported in Table 1 (for Qwen2.5-VL) show a large discrepancy compared to the original paper. For instance, LVBench should report 45.3 for Qwen2.5-VL-7B. This inconsistency raises concerns about the results’ reliability. Although the relative improvement over the baseline is significant, the absolute performance values are not aligned with prior reports.

---

> ### Author Response · Authors · 2025-11-27
> **thanks to the kind reviewer**
>
> We would like to sincerely thank the reviewer for your thoughtful feedback, constructive suggestions, and encouraging comments. Your insights have helped us improve the clarity and completeness of our work, and we deeply appreciate the time and effort you dedicated to evaluating our submission.
>
> Thank you again for your professionalism and fairness throughout the review process. We wish you continued success in your academic careers and hope our paths will cross again in future research endeavors.

---

### Official Review · Reviewer_QhJ1 · 2025-11-03

**Soundness:** 3
**Presentation:** 3
**Contribution:** 3
**Rating:** 8
**Confidence:** 3

**Summary:**

This paper proposes a novel framework, Video-EM, to improve the performance of video QA tasks of long-form video understanding via generating clip-level descriptions and scene details of key events as episodic memory representations and answering on VLMs with them.
The key components to get episodic memory representations are to select key frames, build events via expanding adjacent frames, generate summaries of the form {when, where, what, which object} on events, and construct scene details of object counts and location relationship. The framework integrates Chain-of-Thought (CoT) reasoning on VLMs with them.
It seems that the effectiveness on the proposed methods are validated experimentally with the state-of-the-art results on 4 long-video understanding benchmarks.

**Strengths:**

- The paper identifies the bottlenecks in the previous methods for long-form video understanding, which focus on context window limitations and keyframe redundancy. The proposed idea seem  well-motivated and easy to analyze with readable representation for key event as episodic memories.

- Video-EM looks training-free and integrated with other Video-LLM backbones. It shows good modularity and extensibility.

- The paper provides experimental results on 4 benchmarks (Video-MME, LVBench, HourVideo, Egoschema), consistently outperforming state-of-the-art methods with fewer frames.

**Weaknesses:**

- It seems that the overall performance of Video-EM highly depends on computer vision modules such as object detection, boundary decision and captioning components. In complex or atypical scenes, misdetections or poor captions can undermine the reliability.

- I think it would be helpful to provide failure cases to consider weakness and robustness for the audience.

**Questions:**

- While the modularity of Video-EM is emphasized, the dependencies between modules (e.g., how errors propagate from object detection to CoT reasoning) are not deeply analyzed.
The robustness of the system under suboptimal conditions (e.g., noisy input, failed detection) is not empirically validated, which is crucial for assessing the reliability of the proposed approach.
How does the framework handle errors in object detection or captioning? Are there any mechanisms within somewhere such as CoT reasoning to mitigate or correct such errors?

- I don’t think this manuscript provides enough information to reproduce all of the results. It can be helpful to open the code to resolve this issue. Code will be publicly available?

---

> ### Author Response · Authors · 2025-11-27
> **thanks for the kind reviewer**
>
> We would like to sincerely thank the reviewer for your thoughtful feedback, constructive suggestions, and encouraging comments. Your insights have helped us improve the clarity and completeness of our work, and we deeply appreciate the time and effort you dedicated to evaluating our submission.
>
> Thank you again for your professionalism and fairness throughout the review process. We wish you continued success in your academic careers and hope our paths will cross again in future research endeavors.

---

### Note · Authors · 2026-01-02

I have read and agree with the venue's withdrawal policy on behalf of myself and my co-authors.